# The Association between Non-Alcoholic Fatty Liver Disease (NAFLD) and Advanced Fibrosis with Serological Vitamin B12 Markers: Results from the NHANES 1999–2004

**DOI:** 10.3390/nu14061224

**Published:** 2022-03-14

**Authors:** Li Li, Qi Huang, Linjian Yang, Rui Zhang, Leili Gao, Xueyao Han, Linong Ji, Xiantong Zou

**Affiliations:** Department of Endocrinology and Metabolism, Peking University People’s Hospital, Beijing 100044, China; lili0119@bjmu.edu.cn (L.L.); huangqi16103@pku.edu.cn (Q.H.); yanglj9973@yeah.net (L.Y.); rachelhope@126.com (R.Z.); plum_jj@sina.com (L.G.); xueyaohan@bjmu.edu.cn (X.H.); jiln@bjmu.edu.cn (L.J.)

**Keywords:** vitamin B12, homocysteine, methylmalonic acid, red blood cell folate, folate

## Abstract

Background: There is evidence that vitamin B12 and associated metabolite levels are changed in non-alcoholic fatty liver disease (NAFLD) and non-alcoholic steatohepatitis (NASH); however, their association has been in dispute. Methods: We included 8397 individuals without previous liver condition or excess alcohol intake from the National Health and Nutrition Examination Survey (NHANES) 1999–2004. NAFLD was diagnosed with Fatty Liver Index (FLI) ≥ 60 or USFLI ≥ 30, and participants with advanced fibrosis risks were identified with elevated non-alcoholic fatty liver disease fibrosis score (NFS), fibrosis 4 index (FIB-4), or aspartate aminotransferase (AST)/platelet ratio index (APRI). Step-wide logistic regression adjusting for confounders was used to detect the association between NAFLD or advanced fibrosis with serum vitamin B12, folate, red blood cell folate (RBC folate), homocysteine (HCY), and methylmalonic acid (MMA). Results: The weighted prevalence of NAFLD was 44.2%. Compared with non-NAFLD participants, patients with NAFLD showed significantly increased RBC folate level and RBC counts, decreased serum vitamin B12 and folate, and similar HCY and MMA levels. NAFLD with advanced fibrosis risk had higher MMA and HCY, reduced serum vitamin B12, and similar serum folate and RBC folate levels than NAFLD with low fibrosis risk. Only RBC folate was independently associated with an increased risk of NAFLD (OR (95% CI): 2.24 (1.58, 3.18)). In all participants, MMA (OR: 1.41 (1.10, 1.80)) and HCY (OR: 2.76 (1.49, 5.11)) were independently associated with increased risk for advanced fibrosis. In participants with NAFLD, this independent association still existed (OR: 1.39 (1.04, 1.85) for MMA and 1.95 (1.09, 3.46) for HCY). In all participants, the area under the receiver operating characteristic curve (ROC AUC) on fibrosis was 0.6829 (0.6828, 0.6831) for MMA and 0.7319 (0.7318, 0.7320) for HCY; in participants with NAFLD, the corresponding ROC AUC was 0.6819 (0.6817, 0.6821) for MMA and 0.6926 (0.6925, 0.6928) for HCY. Conclusion: Among vitamin B12-associated biomarkers, RBC folate was independently associated with elevated NAFLD risk, whereas MMA and HCY were associated with increased risk for advanced fibrosis in the total population and NAFLD participants. Our study highlighted the clinical diagnostic value of vitamin B12 metabolites and the possibility that vitamin B12 metabolism could be a therapeutic target for NASH. Further studies using recent perspective data with biopsy proven NASH could be conducted to validate our results.

## 1. Introduction

Non-alcoholic fatty liver disease (NAFLD), affecting approximately 25% of the global population, is one of the most common liver diseases worldwide [1]. The natural history of NAFLD comprises a large pathological spectrum from simple steatosis to steatohepatitis (NASH) with different degrees of fibrosis and cirrhosis [2,3]. NAFLD is highly related to metabolic disorders, including obesity, diabetes, hypertension, and dyslipidemia, and the primary cause of death among NAFLD patients is cardiovascular disease [4]. There is currently no FDA-approved pharmacotherapy specific to NAFLD, and the primary NAFLD treatment is to improve metabolic sequelae and control cardiovascular risks [5]. In addition, surveillance for advanced fibrosis in NAFLD is critical. There is a growing clinical demand to discover and validate novel biomarkers for NAFLD, especially those to identify and predict advanced fibrosis.

A few studies had found that levels of vitamin B12 and its related biomarkers were altered in participants with NAFLD; however, a meta-analysis suggested that compared with non-NAFLD participants, levels of vitamin B12 itself were unchanged [6]. Since vitamin B12 levels and folate were affected by many factors, such as age, dietary habits, and lifestyle, appropriate intermediary compounds such as homocysteine (HCY) and methylmalonic acid (MMA) were more sensitive markers than serum vitamin B12 and folate in reflecting vitamin B12 homeostasis [7]. Red blood cell (RBC) folate represented the average folate levels in the preceding 120 days (the half-life of a RBC), and therefore it was used as an index for accurate tissue folate level [7]. It was found that HCY levels were increased in participants with NAFLD [6]; however, the change in serum HCY in participants with NASH is controversial [8,9,10]. Previous studies had limited sample sizes since the primary diagnostic tool for NASH was liver biopsy, which was less accepted than non-invasive diagnostic tools in patients. To date, non-invasive tests based on serological measurements, such as fatty liver index (FLI), non-alcoholic fatty liver disease fibrosis score (NFS), fibrosis 4 index (FIB-4), and aspartate aminotransferase (AST)/platelet ratio index (APRI) [11,12] have been clinically accepted as surrogate markers to stratify NAFLD and liver fibrosis risks. In this study, we aimed to explore the association between biomarkers related to vitamin B12 metabolism and NAFLD and advanced fibrosis diagnosed by serological non-invasive tests in an extensive epidemiological survey NHANES 1999–2004, which measured multiple vitamin B12 related biomarkers including HCY and MMA.

## 2. Materials and Methods

### 2.1. Study Design and Participants

The National Health and Nutrition Examination Survey (NHANES) is a national, cross-sectional, multistage, probability sampling survey that provides representative samples of the non-institutionalised U.S. resident population (Available online: https://www.cdc.gov/nchs/nhanes/ (accessed on 1 November 2021)). We screened 15,332 participants who were ≥20 years old from NHANES 1999 to 2004 who had blood tested with vitamin B12-related biomarkers including both HCY and MMA. We used the FLI [13] and U.S. Fatty Liver Index (USFLI) [14] for NAFLD diagnosis in this study due to the lack of abdominal ultrasound data. Exclusion criteria included: (1) missing components on FLI or USFLI calculation (*n* = 2642); (2) other existing liver conditions or other causes of chronic liver disease, including hepatitis caused by HBV or HCV and iron overload according to questionnaires and laboratory tests (*n* = 437); (3) excessive alcohol consumption based on questionnaires (*n* = 3711); and (4) missing values on vitamin B12 related biomarkers HCY or MMA (*n* = 145). After exclusion, 8,397 participants were included in our final analysis (Figure 1).

### 2.2. Questionnaire and Lab Tests

General demographic characteristics, including age, sex, race (non-Hispanic White, non-Hispanic Black, Mexican American, or other race), family income, education (<high school, high school and ≥college), physical activity, smoking, and alcohol consumption were collected from self-reported questionnaires collected by interviewing at mobile examination centres (MECs). Family income levels were defined by the poverty income ratio (PIR) as low (PIR < 1.3), middle (PIR 1.3–3.5), and high (PIR > 3.5) [15]. Physical activity was classified as inactivating, moderate, and vigorous on the basis of the activities participants performed over the past 30 days [16]. Dietary folate and vitamin B12 were estimated by calculating the nutrient intake from the answers to the MEC questionnaires. The information of participants taking one or more supplements or drugs containing vitamin B12 or folate in the recent 30 days was collected from the NHANES Dietary Data section and Questionnaire section.

Specimens including blood and urine were collected following standardised procedures. All the detection protocols have been described in detail on the NHANES website (https://www.cdc.gov/nchs/nhanes/) (accessed on 1 November 2021). Briefly, fasting serum biochemistry profiles, including aspartate aminotransferase (AST), alanine aminotransferase (ALT), gamma-glutamyltransferase (GGT), lipids, glucose, and creatinine, were measured using a Hitachi 704 Analyser (Roche/Boehringer Mannheim Corporation, Indianapolis, IN, USA). Blood cell counts were measured using the Beckman Coulter MAXM (Beckman Coulter Corporation, Miami, FL, USA). MMA in plasma or serum was measured using gas chromatography/mass spectrophotometry, and HCY in plasma was measured using a HCY assay kit (Abbott Diagnostics, Abbott Park, IL, USA). Folate (serum and whole blood) and vitamin B12 (serum) levels were measured using the Bio–Rad Laboratories “Quantaphase II Folate/Vitamin B12” radio-assay kit (Bio-Rad Laboratories, Anaheim, CA, USA). Ferritin was tested using the Bio-Rad method in NHANES 1999–2003 and immunoturbidimetry by Roche kits on the Hitachi clinical analyser (Roche Diagnostics, Indianapolis, IN, USA) in 2004. Since the Hitachi method yielded higher ferritin results than the Bio-Rad method, ferritin data were adjusted by linear regression according to analytic notes on the website.

### 2.3. Definitions and Index Calculation

FLI and USFLI were calculated using the following formula as previously described:

FLI = (e^0.953∗ln (TG)+0.139∗BMI+0.718∗ln (GGT)+0.053∗waist circumference−15.745^)/(1 + e^0.953∗ln (TG)+0.139∗BMI+0.718∗ln (GGT)+0.053∗waist circumference−15.745^) ∗ 100 [13].

USFLI = e^(0.3458∗Mexican American−0.8073∗non-Hispanicblack+0.0093∗age+0.6151∗lnGGT+0.0249∗waist circumference+1.1792∗lninsulin+0.8242∗lnGlucose−14.7812)^/(1 + e^(0.3458∗Mexican American−0.8073∗non-Hispanicblack+0.0093∗age+0.6151∗lnGGT+0.0249∗waist circumference+1.1792∗lninsulin+0.8242∗lnGlucos−14.7812)^) ∗ 100 (“non-Hispanic black” and “Mexican American” have a value of 1 if the participant is of that ethnicity and 0 if not of that ethnicity) [14].

Subjects were defined as NAFLD if their FLI score ≥ 60 or USFLI score ≥ 30 in the absence of (1) hepatitis B (positive hepatitis B surface antigen) or hepatitis C infection (positive hepatitis C antibody or HCV RNA); (2) excessive alcohol consumption (>1 alcoholic drink/day for women or >2 alcoholic drinks/day for men) [17]; (3) iron overload, defined as transferrin saturation ≥ 45% along with ferritin ≥ 400 µg/L in women and ≥500 µg/L in men [17]. Serum transferrin saturation was calculated as [serum iron (μmol/L)/serum total iron-binding capacity (μmol/L)] × 100 [18,19].

Advanced fibrosis was assessed by serological non-invasive fibrosis index, including the NFS score, FIB-4, and APRI, calculated using the following formula:NFS = −1.675 + 0.037 ∗ age (years) + 0.094 ∗ BMI (kg/m^2^) + 1.13 ∗ IFG/diabetes (yes = 1, no = 0) + 0.99 ∗ AST/ALT ratio−0.013 ∗ Platelet counts − 0.66 ∗ albumin (g/dL)
FIB-4 = (age ∗ AST)/(Platelet counts ∗ (SQRT(ALT)))
APRI = ([AST/ULN]/Platelet counts) ∗ 100

(AST = 40 U/L was used as the upper limit of normal (ULN) for the 1999–2000 cycle, and 33 U/L was used as the ULN thereafter) [20].

Participants with APRI > 1 or FIB-4 > 2.67 or NFS > 0.676 were regarded as high-risk for advanced fibrosis [21]. NAFLD diagnosed by FLI or USFLI alone and advanced liver fibrosis diagnosed by NFS, FIB-4, and APRI individually were used for sensitivity analysis. Diabetes was defined as a self-report history of diabetes mellitus, fasting glucose levels (FPG) ≥ 7.0 mmol/L, 2 h post-load glucose levels ≥ 11.0 mmol, HbA1c ≥ 6.5% or taking anti-diabetic drugs. Hypertension was defined as blood pressure ≥ 140/90 mmHg or taking anti-hypertension drugs. Dyscholesterolemia was defined as serum total cholesterol (TC) > 200 mg/dL, low-density lipoprotein cholesterol (LDL) ≥ 130 mg/dL, or high-density lipoprotein cholesterol (HDL) < 40 mg/dL for men and <50 mg/dL for women. The estimated glomerular filtration rate (eGFR) was calculated using the equation of Chronic Kidney Disease Epidemiology Collaboration (CKD-EPI). Kidney function was categorised by eGFR as normal (≥90), mild to moderate impairment (60–90), moderate to severe impairment (30–60), and severely decreased kidney function (<30) (mL/min/1.73 m^2^) [22].

Cardiovascular disease (CVD) was defined by self-reported medical history of congestive heart failure, coronary artery disease, heart attack, or stroke. Chronic obstructive pulmonary disease (COPD) was defined by a self-reported medical history of chronic bronchitis or emphysema. Cancer was defined as a self-reported history of any cancer [23].

### 2.4. Statistical Analysis

All statistical analyses were performed in Stata (version 15.0) and R (version 4.1.1) software (“survey” packages in R account for the complex survey design were used). Sampling weights were appropriately adjusted when NHANES 1999–2004 were combined on the basis of the NHANES Analytic and Reporting Guidelines (https://wwwn.cdc.gov/nchs/nhanes/analyticguidelines.aspx#analytic-guidelines) (accessed on 1 November 2021). All estimates were weighted after taking the primary sampling unit, pseudo-strata, and sampling weights to account for the complex sampling design unless otherwise specified.

The data were expressed as percentages for categorical variables and as the mean (95% confidence interval, CI) for continuous variables. For parameters that were not normally distributed, geometric means (95% CIs) were presented. Continuous variables of the participants between the NAFLD versus non-NAFLD groups or fibrosis versus non-fibrosis were estimated using ANCOVA models adjusted by age and sex. Chi-squared test was used to compare categorical data between groups. Logistic regressions were used to investigate the association of MMA or HCY with NAFLD or fibrosis after step-wise adjustment for confounders such as age, sex, race, metabolic disorders, kidney function, serum vitamin B12 and folate level, and history of taking supplement or drugs containing vitamin B12 or folate as specified.

The diagnostic accuracy of MMA or HCY in detecting advanced fibrosis was assessed by weighed receiver operating characteristic curve and the area under the curve (ROC AUC) in different groups and was compared using C statistics. The optimal cut-off point of MMA or HCY was determined by the Youden Index, which was calculated as maximum (sensitivity + specificity − 1).

## 3. Results

### 3.1. Characteristics of Participants with NAFLD and Advanced Fibrosis

After excluding participants with excessive alcohol intake, viral hepatitis, iron overload, and other previous liver conditions and those who missed essential parameters, we identified 8397 participants in this study (detailed inclusion algorithm in Figure 1). The weighted prevalence of NAFLD was 44.2%. Compared to participants without NAFLD, participants with NAFLD had a higher prevalence of metabolic disorders including diabetes, obesity, hypertension, and dyscholestrolemia; more elevated hepatic inflammatory markers including ferritin, CRP, AST, ALT, and GGT; higher RBC counts; and RDW and lower dietary folate. There was a decrease in serum vitamin B12 and folate levels, an increase in RBC folate level, and no difference in MMA and HCY levels in NAFLD participants compared with non-NAFLD participants (Table 1).

The percentage of participants with advanced fibrosis risk was 2.6% in non-NAFLD and 7.0% in NAFLD participants. Compared with NAFLD with low fibrosis risk, NAFLD participants with advanced fibrosis risk had a higher prevalence of obesity, diabetes, hypertension, and CVD, but a lower prevalence of dyscholestrolemia. There was no difference in serum folate or RBC folate between the two groups; however, serum vitamin B12, dietary vitamin B12, and dietary folate were reduced, and MMA and HCY were significantly increased in participants with advanced fibrosis risks (Table 1).

### 3.2. Association between Vitamin B12 Markers and NAFLD or Advanced Fibrosis

There was no association between serum folate and serum vitamin B12 with NAFLD (HR 95% CI: 0.92 (0.75, 1.13) (*p* = 0.425) and 0.90 (0.70, 1.16) (*p* = 0.415), respectively) or advanced fibrosis in all participants (HR 95% CI: 0.94 (0.79, 1.11) (*p* = 0.422) and 1.04 (0.77, 1.40) (*p* = 0.786), respectively) when age, sex, race, smoking, BMI, diabetes, hypertension, dyscholestrolemia, kidney function, and taking vitamin B12 or folate supplement were controlled for. RBC folate was associated with an increased risk for NAFLD (HR 2.46 (1.72, 3.53), *p* < 0.001) when all confounders were adjusted. MMA and HCY were both associated with a crude increase in NAFLD risk; however, when age and sex were corrected, the association became non-significant. In all participants and NAFLD patients, both MMA and HCY were positively associated with an elevated risk of advanced fibrosis when all confounders were adjusted for. In NAFLD, the crude OR for advanced fibrosis was 3.08 (2.40, 3.95) for MMA and 6.81 (4.63, 10.01) for HCY, whereas the adjusted values were 1.39 (1.04, 1.85) for MMA and 1.95 (1.09, 3.46) for HCY. There was no significant association between RBC folate and advanced fibrosis (Table 2).

By sensitivity analysis, we found RBC folate was independently associated with NAFLD diagnosed by either FLI or USFLI alone or in combination. MMA was independently associated with fibrosis defined by NFS in the total population and NAFLD participants, whereas HCY was independently associated with fibrosis defined by FIB-4 or APRI in the total population (Appendix A).

### 3.3. Diagnostic Performance of MMA and HCY for Advanced Fibrosis in NAFLD

The prevalence of elevated fibrosis risk was significantly higher in participants with abnormal MMA and HCY levels (Appendix A). The ROC AUCs (95%CI) of HCY, MMA, ferritin, and CRP on fibrosis were 0.7319 (0.7318, 0.7320), 0.6829 (0.6828, 0.6831), 0.5332 (0.5330, 0.5333), and 0.5995 (0.5994, 0.5996) (*p* < 0.001) in all participants and 0.6926 (0.6925, 0.6928) and 0.6819 (0.6817, 0.6821), 0.5295 (0.5293, 0.5297), and 0.5229 (0.5227, 0.5231) in NAFLD participants, respectively (Figure 2). The optimal cut-off values were 178 nmol/L and 179 nmol/L of MMA and 9.3 umol/L and 10.0 umol/L of HCY to detect advanced fibrosis in all and in NAFLD participants, respectively.

## 4. Discussion

This study demonstrated that NAFLD participants had increased RBC folate levels and decreased serum vitamin B12, as well as unchanged MMA and HCY compared with non-NAFLD controls. NAFLD participants with advanced liver fibrosis risk had higher HCY and MMA levels but similar serum folate and RBC folate levels compared to those with low fibrosis risk. RBC folate was independently associated with NAFLD, whereas HCY and MMA were independently associated with advanced fibrosis risk in NAFLD participants.

We established the associations of biomarkers in vitamin B12 metabolism, including RBC folate, HCY, and MMA, with NAFLD or advanced fibrosis risks. Serum vitamin B12 and folate levels were not associated with NAFLD or advanced fibrosis. Consistently, a meta-analysis showed that vitamin B12 levels were similar between participants with and without NAFLD [6]. Serum vitamin B12 and folate were affected by many factors, such as age, dietary habits, and lifestyle. HCY and MMA were more sensitive markers than serum vitamin B12 and folate to reflect vitamin B12 homeostasis [7], especially when vitamin B12 levels were above the lower normal limit. Folate in RBC is more stable than serum folate [7]. Although there were a few studies that investigated RBC folate and NAFLD, previous studies suggested that RBC folate was associated with insulin resistance and metabolic syndrome [24]. In addition, our data suggested that RBC folate was associated with diabetes (Appendix A), illuminating the role of RBC folate in metabolic diseases.

Our study suggested that HCY was independently associated with the risk of advanced fibrosis in NAFLD. It was demonstrated that HCY was increased in participants with histology-confirmed NAFLD [6,10]. In a sizeable Chinese epidemiological study, HCY was associated with NAFLD diagnosed by ultrasound [25]. In contrast with prior studies, our study suggested that HCY was not independently associated with NAFLD. This could possibly have been due to the different ethnicities and the diagnostic tools used for NAFLD. The association between HCY and NASH or advanced fibrosis was less studied and was still in dispute. In histologically confirmed NASH, it was found that HCY was negatively [8,10], positively [9], or unrelated [26] to the severity of NASH or the stages of fibrosis. Ethnic differences, sample sizes and different diagnostic criteria may explain the discrepancy. Our study provided epidemiological data in a large sample size to support that HCY was positively associated with advanced fibrosis risks in NAFLD.

We identified a novel biomarker, MMA, independently related to advanced fibrosis in total population and in NAFLD. MMA, an intermediate mitochondrial metabolite of odd-chain fatty acids and catabolism of several amino acids, is converted into succinic acid and enters the Krebs cycle under normal conditions. This process is catalysed by mitochondrial methylmalonyl-CoA mutase (MMUT) and coenzyme vitamin B12 so that vitamin B12 deficiency leads to MMA accumulation. MMA is a sensitive marker for vitamin B12 deficiency [27]. Increasing evidence has supported the fact that MMA is not only a metabolite indicating vitamin B12 deficiency, but also a marker for mitochondrial dysfunction and oxidative stress in vivo and in vitro [28]. In NHANES 1999–2004, MMA independently predicted all-cause and cause-specific mortality [29]. MMA levels were also associated with insulin resistance and metabolic syndrome [24,30], and this was confirmed with our study that MMA was associated with the risk of hypertension (Appendix A). Nevertheless, the relationship between MMA and NAFLD or NASH has not been investigated previously. Our study suggested that MMA is associated with NAFLD at a crude level; however, after all confounders were adjusted for, MMA was not an independent predictor for NALFD.

The role of HCY and MMA in advanced fibrosis involves one-carbon metabolism, the network of interrelated biochemical reactions that involve the transfer of one-carbon (methyl) groups from one compound to another [7]. It was found that during the progression from steatosis to steatohepatitis, dysregulation of the one-carbon metabolism occurs, involving metabolites such as HCY, methionine, and betaine [31]. HCY acted as an essential intermediate metabolite during the one-carbon metabolism and acted as a crucial indicator of redox homeostasis [32,33]. It is independently associated with stroke, hypertension, and other cardiovascular risks [34]. Our study indicated a possible link between on-carbon metabolites and advanced liver fibrosis in NAFLD. Further studies were inspired by a search for metabolites or the combination of metabolites in the one-carbon metabolism as predictors for advanced fibrosis.

The mechanism of the association between HCY and MMA and advanced fibrosis also involves mitochondrial dysregulation. Mitochondrial lipid beta-oxidation, oxidative stress, and mitophagy all played important roles in steatosis and the progression from steatosis to NASH [35]. HCY was increased in hepatic steatosis induced by the inhibition of mitochondrial fatty acid oxidation, suggesting a possible role of HCY in hepatic lipid metabolism [36]. HCY could induce steatosis by increasing de novo lipogenesis through the upregulation of ER stress [37]. In Wistar rats, MMA was elevated in diet-induced hepatic steatosis and was identified as a novel circulating marker reflecting mitochondrial β-oxidation [38]. A recent study found that MMA accumulation induced by MMUT deficiency was involved in metabolic and mitochondrial alterations that were exacerbated by anomalies in PINK1/Parkin-mediated mitophagy, causing the accumulation of dysfunctional mitochondria and triggering epithelial stress and ultimately cell damage [39].

In the clinical setting, our study provides insights into the diagnostic value of HCY and MMA to detect advanced fibrosis in NAFLD participants. Identifying the risk for advanced fibrosis is an essential step in NASH management. To date, the diagnosis primarily relies on liver biopsy, and numerous non-invasive techniques have been developed as substitutions. The ROC AUC of HCY and MMA can hardly meet the standard as clinical biomarkers; however, they performed better than known predictors such as ferritin [40] and CRP [41]. The actual diagnostic performance needs to be further validated in outcomes diagnosed by ultrasound or biopsy. Nevertheless, we highlighted the possibility that metabolites involved in vitamin B12 metabolism and homeostasis can be used alone or in combination to predict advanced fibrosis in NASH. Our study also indicated the possibility of treating HCY and MMA as therapeutic targets for NASH. It was found that methyl-donor supplementation, which could lower HCY levels, could prevent NAFLD progression [42]. Folate supplementation, a common method of treating hyperhomocysteinemia, was found to prevent high fructose-induced NAFLD by activating the AMPK and LKB1 signalling pathways [43]. Further clinical trials on humans can be considered when more evidence is accumulated.

One strength of our study was that we expanded the sample size using well-established indexes derived from basic anthropometric parameters and serum markers to diagnose steatosis and fibrosis in NAFLD. The FLI had an AUC of 0.84, and the USFLI had an AUC of 0.80 to detect steatosis in the U.S. population [14]. We used a combination of these two indexes as previously described [44]. We performed a sensitivity analysis to test these patients and found that the association between RBC folate and NAFLD was stable whether FLI or USFLI was used (Appendix A). We used the combination of three non-invasive serological fibrosis indexes to detect the risk for advanced fibrosis as previously described. The NFS, APRI, and FIB-4 values had AUCs of 0.81 [45], 0.73 [46], and 0.80 [46], respectively, in the detection of advanced fibrosis when they were first developed or validated in NAFLD.

Our study had several limitations: (1) We did not use the gold standard to diagnose NAFLD or NASH, and we did not include imaging for NAFLD and NASH diagnosis because of the lack of data. The results, especially the association between MMA and advanced fibrosis, need to be validated in cohorts with biopsy information. (2) Another major limitation was that the cohort and tests were conducted almost 20 years ago. During this period, laboratory techniques, management, the spectrum of co-morbid diseases, and dietary habit of people have changed. However, only NHANES 1999–2004 tested both HCY and MMA, and therefore we cannot collect data from other NHANES cohorts due to unavailability. We adjusted as many confounders as possible to prove the independent associations between HCY or MMA and fibrosis. However, validation should be conducted in recent studies and studies from different populations. As a cross-sectional study was not able to provide a causal relationship between HCY and MMA and NASH, further studies in prospective cohorts are necessary. (3) We were unable to test other metabolites involved in vitamin B12 metabolism, and we were unable to adjust dietary habits due to the lack of data.

## 5. Conclusions

Among biomarkers related to vitamin B12, RBC folate was independently associated with NAFLD, and MMA and HCY were independently associated with advanced fibrosis risks in NAFLD, suggesting a possible role of vitamin B12 homeostasis in the pathogenesis of NASH. Our study provided evidence that HCY and MMA could be candidate biomarkers to predict fibrosis in NAFLD and inspired relative therapeutic strategies.

## Figures and Tables

**Figure 1 nutrients-14-01224-f001:**
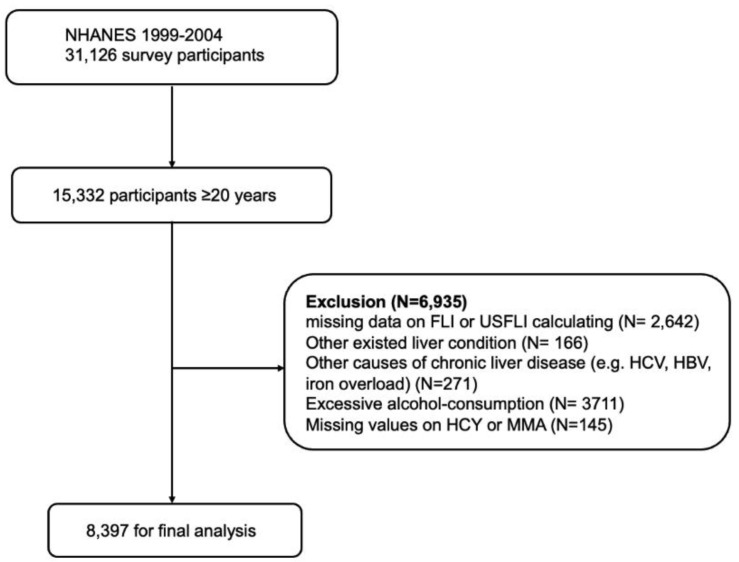
Inclusion algorithm.

**Figure 2 nutrients-14-01224-f002:**
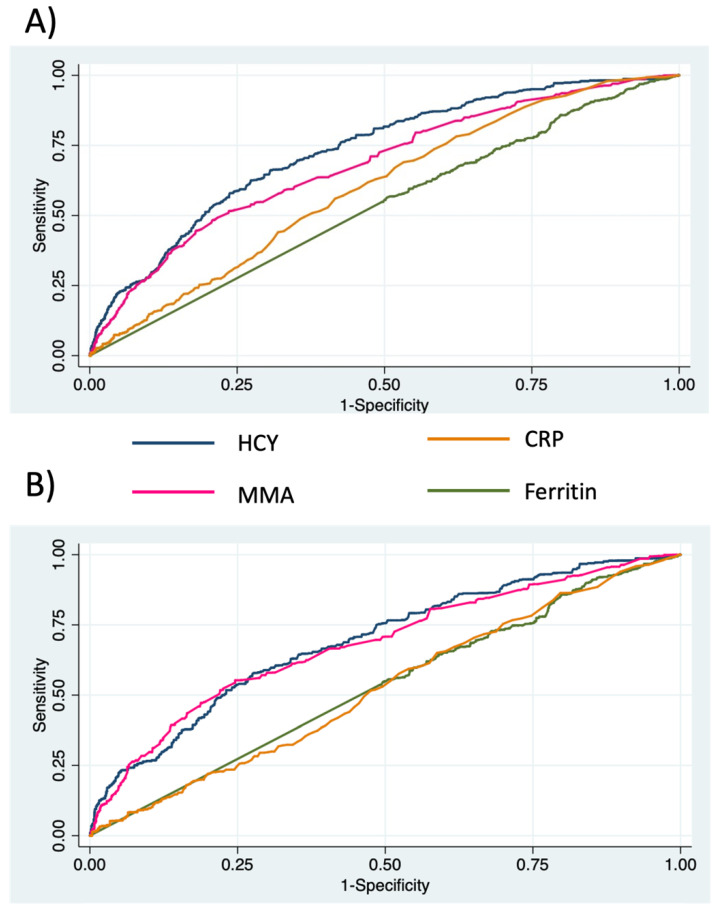
Receiver operator curve (ROC) of biomarkers in all participants (**A**) and participants with NAFLD (**B**). MMA, methylmalonic acid; HCY, homocysteine. CRP: c-reactive protein.

**Table 1 nutrients-14-01224-t001:** Baseline characteristics of participants in NHANES 1999–2004.

		Overall	*p*	Non-NAFLD	NAFLD			
			(NAFLD vs. Non-NAFLD)		Total	Low Fibrosis Risk	High Fibrosis Risk	*p* (High vs. Low Fibrosis Risk)
	*n* (%)	8397 (100%)		4562 (55.8)	3835 (44.2)	3475 (93.0)	360 (7.0)	
NAFLD and fibrosis scores	FLI $	50.2 (49.0, 51.4)	<0.001	26.3 (25.8, 26.8)	81.5 (80.7, 82.4)	80.7 (80.0, 81.4)	88.8 (86.3, 91.4)	<0.001
	USFLI $	23.6 (22.4, 24.9)	<0.001	10.9 (10.5, 11.3)	40.5 (38.9, 42.1)	39.6 (38.1, 41.1)	54.9 (50.2, 59.6)	<0.001
	NFS $	−2.24 (−2.29, −2.19)	<0.001	−2.35 (−2.39, −2.32)	−1.67 (−1.73, −1.61)	−1.72 (−1.77, −1.67)	0.40 (0.24, 0.56)	<0.001
	FIB-4 $	1.02 (1.00, 1.04)	<0.001	1.17 (1.16, 1.19)	1.04 (1.02, 1.05)	1.03 (1.02, 1.05)	1.84 (1.72, 1.96)	<0.001
	APRI $	0.287 (0.283, 0.291)	0.3080	0.293 (0.287, 0.299)	0.288 (0.283, 0.294)	0.271 (0.265, 0.276)	0.507 (0.462, 0.552)	<0.001
Demographic data	Ethnicity (%)		<0.001					<0.001
	Non-Hispanic Black	10.8		11.0	10.6	10.6	10.5	
	Mexican American	6.3		6.4	6.1	6.3	4.4	
	Other race	9.9		11.0	8.7	8.8	6.9	
	Income level $, %		0.4469					<0.001
	Low	20.2		19.9	20.5	19.8	30.5	
	Middle	36.3		35.8	37.1	36.8	41.1	
	High	43.5		44.3	42.5	43.5	28.3	
	Education level $, %		<0.001					0.0024
	<High school	19.7		18.3	21.4	20.7	31.1	
	High school	25.6		24.0	27.6	27.7	27.0	
	≥college	54.8		57.8	51.0	51.6	41.9	
	Physical activity $, %		<0.001					<0.001
	Inactivate	36.9		32.6	42.5	41.3	58.0	
	Moderate	31.9		30.9	33.1	33.2	31.5	
	Vigorous	31.2		36.6	24.4	25.4	10.5	
	Ever smoking, %	42.7	<0.001	38.8	47.6	47.1	54.0	0.0561
Dietary	Diet folate $, mcg	402 (391, 412)	<0.001	413 (401, 426)	384 (374, 394)	384 (374, 395)	352 (326, 377)	0.0299
	Diet vitamin B12 $, mcg	5.22 (4.93, 5.51)	0.1149	5.09 (4.76, 5.43)	5.37 (5.09, 5.64)	5.42 (5.15, 5.69)	4.47 (3.82, 5.12)	0.0097
	Taking vitamin B12 or folate supplement, %	22.6	0.0318	23.6	21.4	21.4	21.0	0.8893
Physical examinations	BMI $, kg/m^2^	28.4 (28.1, 28.6)	<0.001	24.4 (24.2, 24.5)	33.2 (32.8, 33.5)	32.6 (32.3, 32.8)	37.9 (36.3, 39.5)	<0.001
	Waist circumference, cm	97.6 (96.9, 98.3)	<0.001	88.3 (87.9, 88.8)	110.0 (109.2, 110.8)	109.0 (109.0, 110.0)	119.0 (116.0, 122.0)	<0.001
	SBP $, mmHg	124.2 (123.3, 125.0)	<0.001	124.0 (123.0, 125.0)	129.0 (128.0, 130.0)	130.0 (129.0, 131.0)	129.0 (126.0, 132.0)	0.7686
	DBP $, mmHg	71.6 (71.1, 72.0)	<0.001	69.8 (69.4, 70.3)	73.3 (72.6, 74.0)	73.1 (72.4, 73.8)	69.2 (67.9, 70.6)	<0.001
Biochemical Laboratory Tests	Glucose $, mmol/L	5.67 (5.59, 5.75)	<0.001	5.40 (5.35, 5.44)	6.17 (6.06, 6.29)	6.17 (6.05, 6.29)	7.19 (6.67, 7.70)	<0.001
	Total cholesterol $, mmol/L	5.24 (5.20, 5.28)	<0.001	5.14 (5.11, 5.18)	5.45 (5.40, 5.51)	5.49 (5.43, 5.55)	5.01 (4.87, 5.14)	<0.001
	TG$ #, mmol/L	1.32 (1.29, 1.35)	<0.001	1.04 (1.02, 1.07)	1.84 (1.79, 1.89)	1.85 (1.80, 1.90)	1.69 (1.55, 1.84)	0.0322
	HDL $, mmol/L	1.34 (1.33, 1.36)	<0.001	1.47 (1.45, 1.48)	1.20 (1.18, 1.22)	1.20 (1.19, 1.22)	1.21 (1.15, 1.26)	0.8820
	ALT, U/L	24.9 (24.2, 25.5)	<0.001	21.3 (20.8, 21.7)	28.8 (27.4, 30.1)	27.5 (26.2, 28.7)	34.8 (30.5, 39.1)	0.0021
	AST, U/L	24.3 (24.0, 24.5)	<0.001	23.8 (23.6, 24.1)	25.0 (24.7, 25.4)	24.2 (23.8, 24.5)	34.0 (30.2, 37.9)	<0.001
	GGT, U/L	26.0 (25.2, 26.9)	<0.001	19.7 (19.2, 20.3)	34.6 (33.1, 36.2)	33.9 (32.6, 35.1)	43.2 (35.6, 50.9)	0.0166
	CRP $, mg/dL	0.43 (0.41, 0.46)	<0.001	0.29 (0.27, 0.31)	0.61 (0.56, 0.66)	0.61 (0.56, 0.65)	0.68 (0.55, 0.80)	0.2187
	eGFR, mL/min per 1.73 m^2^	92.5 (91.3, 93.7)	0.6370	89.3 (88.3, 90.3)	89.0 (88.0, 90.0)	87.0 (86.0, 88.0)	82.0 (79.3, 84.7)	<0.001
	Ferritin $, μg/L	123 (118, 128)	<0.001	115 (110, 120)	148 (141, 155)	147 (140, 155)	180 (146, 213)	0.0816
Vitamin B12 markers	MMA #, nmol/L	139.5 (135.7, 143.4)	0.4490	145.5 (141.2, 148.4)	144.0 (139.8, 146.9)	145.5 (141.2, 149.9)	174.2 (164.0, 184.9)	<0.001
	HCY #, μmol/L	8.30 (8.12, 8.48)	0.1880	8.58 (8.41, 8,76)	8.67 (8.50, 8.85)	8.76 (8.58, 8.94)	9.78 (9.21, 10.49)	0.0011
	Folate $#, serum (nmol/L)	29.99 (29.15, 30.86)	<0.001	32.46 (31.19, 33.45)	29.37 (28.50, 29.96)	29.96 (29.08, 31.19)	29.67 (27.66, 31.82)	0.7216
	Folate $#, RBC (nmol/L RBC)	648.5 (633.3, 663.9)	<0.001	649.4 (632.1, 667.1)	679.3 (663.8, 695.1)	690.2 (673.8, 707.0)	695.1 (659.8, 732.2)	0.7940
	Vitamin B12 $#, pmol/L	347.5 (339.4, 355.8)	<0.001	364.3 (353.9, 375.0)	330.6 (324.1, 337.0)	333.0 (326.4, 340.0)	306.7 (289.7, 325.1)	0.0112
Red blood cell markers	RBC $ (million cells/μL)	4.74 (4.71, 4.76)	<0.001	4.67 (4.65, 4.70)	4.80 (4.77, 4.82)	4.80 (4.77, 4.82)	4.64 (4.58, 4.70)	<0.001
	RDW $ (%)	12.67 (12.63, 12.70)	<0.001	12.61 (12.57, 12.65)	12.80 (12.75, 12.85)	12.79 (12.74, 12.85)	13.19 (13.01, 13.37)	<0.001
	MCV $ (fL)	90.24 (90.00, 90.47)	<0.001	91.15 (90.92, 91.38)	89.74 (89.47, 90.01)	89.94 (89.67, 90.20)	90.23 (89.53, 90.94)	0.3930
	MCHC $ (g/dL)	33.84 (33.77, 33.91)	0.0029	33.80 (33.73, 33.86)	33.88 (33.80, 33.95)	33.87 (33.79, 33.94)	33.89 (33.81, 33.98)	0.5588
Metabolic disorders	Obesity, %	32.3	<0.001	5.1	66.8	65.8	80.0	<0.001
	Hypertension, %	39.5	<0.001	28.3	53.6	51.6	80.7	<0.001
	Diabetes, %	11.2	<0.001	4.4	19.7	16.6	60.2	<0.001
	Dyscholestrolemia, %	68.5	<0.001	58.8	80.9	81.6	71.5	<0.001
Comorbidities	CVD, %	10.7	<0.001	7.9	14.3	12.4	39.4	<0.001
	COPD, %	4.4	0.0304	3.7	5.3	4.9	11.0	<0.001
	Cancer, %	9.3	0.3014	9.0	9.7	9.0	18.6	<0.001

For continuous variables, data were presented as weighed mean (95% CI). The p-value was estimated with the ANCOVA model adjusted by age and gender when appropriate for continuous variables. Data without normal distribution were log-transformed for statistical analysis, and geometric mean (95%CI) was presented (#). For categorical variables, data were presented as weighed proportion (%), and significance was tested using the chi-squared test. $: With missing value. NAFLD, non-alcoholic fatty liver disease; FLI, Fatty Liver Index; USFLI, U.S. Fatty Liver Index; NFS, NAFLD fibrosis score; FIB-4, Fibrosis-4 index; APRI, AST-to-platelet ratio index; BMI, body mass index; SBP, systolic blood pressure; DBP, diastolic blood pressure; TG, triglyceride; HDL, high-density lipoprotein cholesterol; ALT, alanine aminotransferase; AST, aspartate transaminase; GGT, gamma-glutamyl transferase; CRP, C-reactive protein; eGFR, estimated glomerular filtration rate; MMA, methylmalonic acid; HCY, homocysteine; RBC, red blood cell; RDW, red blood cell distribution width; MCV, mean cell volume; MCHC, mean cell haemoglobin concentration; CVD, cardiovascular disease; COPD, chronic obstructive pulmonary disease; CI, confidence interval.

**Table 2 nutrients-14-01224-t002:** The association of MMA, HCY, and RBC-folate with NAFLD or advanced fibrosis in participants from NHANES 1999–2004.

			Advanced Fibrosis	
	(OR)	NAFLD	Total	NAFLD
Model 1	MMA	1.21 (1.10, 1.34) **	2.86 (2.43, 3.38) **	3.08 (2.40, 3.95) **
	HCY	1.95 (1.66, 2.29) **	7.86 (6.08, 10.16) **	6.81 (4.63, 10.01) **
	RBC-folate	1.56 (1.32, 1.84) **	2.32 (1.77, 3.04) **	2.17 (1.49, 3.16) **
Model 2	MMA	0.95 (0.85, 1.07)	1.60 (1.34, 1.91) **	1.73 (1.38, 2.16) **
	HCY	1.12 (0.93, 1.34)	3.03 (1.99, 4.62) **	2.46 (1.55, 3.89) **
	RBC-folate	1.42 (1.20, 1.67) **	1.09 (0.83, 1.44)	1.13 (0.78, 1.63)
Model 3	MMA	1.13 (0.95, 1.34)	1.32 (1.05, 1.66) *	1.36 (1.05, 1.77) *
	HCY	0.84 (0.55, 1.28)	2.29 (1.35, 3.89) *	1.77 (1.07, 2.93) *
	RBC-folate	1.58 (1.20, 2.07) *	1.05 (0.84, 1.31)	1.27 (0.91, 1.77)
Model 4	MMA	1.10 (0.89, 1.36)	1.41 (1.10, 1.80) *	1.39 (1.04, 1.85) *
	HCY	0.75 (0.47, 1.18)	2.76 (1.49, 5.11) *	1.95 (1.09, 3.46) *
	RBC-folate	2.24 (1.58, 3.18) **	1.20 (0.90, 1.60)	1.34 (0.89, 2.01)
Model 5	RBC-folate	2.46 (1.72, 3.53) **	1.18 (0.88, 1.57)	1.33 (0.88, 1.99)

Logistic regression was used to detect the odds ratio (95% CI) for NAFLD and advanced fibrosis of MMA, HCY, and RBC-folate; ** *p* ≤ 0.001, * *p* < 0.05. Model 1: unadjusted; Model 2: age + sex + races + smoking; Model 3: model 2 + BMI + diabetes + hypertension + dyscholestrolemia + kidney function; Model 4: model 3 + log-transformed-vitamin B12 + log-transformed-serum folate + taking vitamin B12 or folate supplement; Model 5: model 4 + haemoglobin; MMA, methylmalonic acid. HCY, homocysteine; RBC-folate: folate in red blood cells; NAFLD, non-alcoholic fatty liver disease.

## Data Availability

Publicly available datasets were analysed in this study. These data can be found at https://www.cdc.gov/nchs/nhanes (accessed on 1 November 2021).

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
