# Peer review of "The Association between Non-Alcoholic Fatty Liver Disease (NAFLD) and Advanced Fibrosis with Serological Vitamin B12 Markers: Results from the NHANES 1999–2004"

_nutrients, 2022, doi:10.3390/nu14061224_

Round 1
Reviewer 1 Report
The authors utilized the survey data, NHANS, to explore the possible associations between NAFLD and liver fibrosis with serum B12 markers. The hypothesis and the analytical procedures in the study are generally practical; however, some information could be clarified for the readers.
- To the best of my knowledge, NHANS is still an active database. Therefore, newer versions of the information should be applicable for the analysis. After reading, I still have no idea why NHANES 1999-2004, more than 20 years ago data, was selected for the analysis. If there are some essential reasons of using this period, the authors must make a clear statement to the readers.
- In case that the 1999-2004 data was selected due to an essential reason, this should be a big limitation of this study which have to be clarified. As authors mentioned that diagnosis, laboratory results, management, co-morbid diseases, and dietary habit of people during more than 20 years ago was extremely changed, what should be the benefits to our medicinal network in the presence.
- According to the B12-related markers in the study, what if those patients in the study took folic acid or vitamin B12 derivatives as a supplements for the specific diseases or conditions? Was there any related information to be clarified?
- It would be nice to briefly explain why red blood cell (RBC) folate, homocysteine (HCY) and methylmalonic acid (MMA) are the biomarkers or vitamin B12.
- The authors concluded that HCY and MMA should be a candidate biomarker for NAFLD. It would be better to suggest if these candidates are compared with others available purposed biomarkers.
- The authors also concluded that targeting vitamin B12 deficiency may be an option for NAFLD management. The question is how people could be diagnosed as B12 deficiency. Are there any cut-off level for this condition relating to study information?
- Some typos can be found, such as in the conclusion: red blood cell (RBC) folate, homocysteine (HCY) and methylmalonic acid (MMA)...
Reviewer 2 Report
The collection of such a serial data is not easy. For this reason, the paper deserves such a credit for carrying out such a study. Most of analyses are carried out in a very reasonable way. I agree with the finding of the results may be important to the disease progression.
Author Response
Thank you very much. Your evaluation was very meaningful to us.
Reviewer 3 Report
1)The introduction could be expanded by enriching the section on biomarkers and their relation to NAFLD.
2)Review the bibliographic references and their placement in the text.
Author Response
Response 1: Sorry for the unclarity of the last version. We have revised the second paragraph of the introduction part according to your suggestion. The need for using alternative biomarkers and their relationships with NAFLD was specified.
Response 2: Thanks for your suggestion. We have checked and updated the reference and their placement of the whole text.
Round 2
Reviewer 1 Report
Is it possible to rewrite/reconstruct results of the abstract? It is pretty unclear due to 1) some long sentences with a lot of conjunctions within 2) difficulty relating statistics and abbreviations (AUC is generally recognized, but not for ROC AUC).
No other points to be addressed.
Author Response
Response: Apologies for the unclarity of the previous version and thank you very much for your suggestion. We have made corrections to the result part of the abstract to make the statistics specific to biomarkers. We also clarified the full name of the abbreviation.